# Relationships between personality, emotional well-being, self-efficacy and weight management among adults with type 2 diabetes: Results from a cross-sectional survey

Ralph Geerling[1,2]*, Jeromy Anglim[1], Emily J. Kothe[1], Miranda T. Schram[3,4], Elizabeth Holmes-Truscott[1,2], Jane Speight[1,2]

**1** School of Psychology, Deakin University, Geelong, Australia, **2** The Australian Centre for Behavioural Research in Diabetes, Diabetes Victoria, Melbourne, Australia, **3** Department of Internal Medicine, School of Cardiovascular Disease (CARIM), Maastricht University, Maastricht, The Netherlands, **4** Heart and Vascular Center, Maastricht University Medical Center+, Maastricht, The Netherlands

* rgeerling@acbrd.org.au

## Abstract

The objective of this study was to examine the associations between personality, general and diabetes-specific well-being and self-efficacy, and weight management indicators, among adults with type 2 diabetes. In addition, to examine whether personality provides incremental explanation of variance in weight management indicators. Australian adults with type 2 diabetes (N = 270; 56% women; age: 61±12 years) were recruited via the national diabetes registry. An online survey included measures of: personality (HEXACO-PI-R), weight management indicators (physical activity, healthy diet, body mass index [BMI]), general well-being (WHO-5), general self-efficacy (GSE), diabetes distress (DDS) and diabetes self-efficacy (DMSES). Analyses included bivariate correlations and linear regression, adjusted for demographic, clinical, and psychological variables. All six personality domains showed significant correlation with at least one weight management indicator: physical activity with extraversion ($r = .28$), conscientiousness ($r = .18$) and openness ($r = .19$); healthy diet with honesty-humility ($r = .19$), extraversion ($r = .24$), and agreeableness ($r = .14$); and BMI with emotionality ($r = .20$) and extraversion ($r = -.20$). The strongest associations with general and diabetes-specific well-being and self-efficacy were apparent for extraversion, emotionality and conscientiousness (range: $r = -.47-.66$). Beyond covariates, personality domains explained additional variance for physical activity (Adjusted $R^2 = .31$, $R^2$ difference = .03, $p = .03$; openness: $\beta = .16$, $p = .02$, emotionality: $\beta = .15$, $p = .04$) and healthy diet (Adjusted $R^2 = .19$, $R^2$ difference = .03, $p = .02$; honesty-humility: $\beta = .20$, $p = .002$, extraversion: $\beta = .19$, $p = .04$) but not BMI. This study shows that personality is associated with weight management indicators and psychological factors among adults with type 2 diabetes. Further research is needed, including objective measurement of weight management indictors, to examine how personality influences the experience of type 2 diabetes.

**Data Availability Statement:** All data files are available from the Open Science Framework database (https://osf.io/ncepq/).

**Funding:** RG is supported by a Deakin University Industry PhD Scholarship, in collaboration with AstraZeneca Australia www.astrazeneca.com.au (unrestricted educational grant). JS and EHT are supported by core funding of The Australian Centre for Behavioural Research in Diabetes (ACBRD) www.acbrd.org.au provided by the collaboration between Diabetes Victoria and Deakin University www.deakin.edu.au. The funders had no role in study design, data collection and analysis, decision to publish, or preparation of the manuscript.

**Competing interests:** The authors have declared that no competing interests exist.

## Introduction

Healthful behaviours such as physical activity and healthy eating are the cornerstone of type 2 diabetes management with positive impacts on both glucose and weight management [1]. However, for most adults with type 2 diabetes, uptake of physical activity, dietary intake, and achieved weight loss, is not consistent with clinical guidelines [1]. Unique challenges also exist for people living with diabetes regarding insulin sensitivity and medication, including insulin, that can affect weight gain [2]. These unique physiological challenges may be compounded by behavioural and psychological consequences, such as demotivation and diabetes distress, creating a reinforcing cycle [3]. Research has examined barriers to, and antecedents of, self-care and weight management among adults with type 2 diabetes, identifying associations with emotional well-being and self-efficacy [4]), as well as with certain personality traits [5]. Research also shows that general well-being and general self-efficacy correlate with diabetes distress and diabetes self-efficacy, respectively [3,6], but their association with underlying personality traits is less well established among adults with type 2 diabetes.

General emotional well-being and self-efficacy are both relatively stable characteristics with substantial overlap with comprehensive personality frameworks [7]; Personality traits—defined as consistent individual differences in how people think, feel and behave [8]—are most commonly represented by the broad and empirically validated trait framework known as the Big Five [8]. The Big Five traits includes openness to experience (e.g. open-minded, intellectual), conscientiousness (e.g. disciplined, orderly), extraversion (e.g. sociable, active), agreeableness (e.g. trusting, caring), and neuroticism (e.g. anxious, stressed) [8]. Meta-analyses show that well-being is most notably correlated with neuroticism (negative), followed by extraversion and conscientiousness (positive) [7]; self-efficacy has somewhat similar personality correlates to well-being but with stronger associations for conscientiousness (positive) [9]. In contrast, less is known about how the traits in comprehensive personality frameworks correlate with diabetes distress and diabetes-specific self-efficacy. Exploring these relationships may enrich our understanding of the complexities of living with type 2 diabetes and managing weight.

Recent research building on the Big Five has demonstrated the utility of a sixth domain: honesty-humility (e.g. fair, modest). This representation of personality is most prominently captured in the HEXACO model (honesty-humility, emotionality, extraversion, agreeableness, conscientiousness and openness to experience), which incorporates the six broad domains each comprising four narrower facets [10]. Whilst the Big Five is the most commonly used, and validated, representation of broad personality traits, the HEXACO model has demonstrated predictive advantages over the Big Five [10]. Further, the honesty-humility domain offers unique insights, for example, into the role of personality in weight management in the general population [11].

Within the obesity population and among people with cardiovascular disease, effective weight management is generally related to higher conscientiousness and lower neuroticism and, less consistently, with extraversion [12–15]. Yet, comparatively fewer studies have specifically examined the personality-weight management relationship in type 2 diabetes. A recent systematic review of 17 studies found none had the direct objective of examining the personality-weight management relationship and none utilised the HEXACO [5]. With weight management being a vital component of diabetes management, it is necessary to specifically and directly investigate the role of personality in weight management in a type 2 diabetes sample. The review [5] also found personality constructs representing negative emotionality (negatively) as well as conscientiousness (positively) were associated with the performance of weight management behaviours and outcomes. Whether these associations can be replicated will be

an important aspect of developing the personality-weight management research in type 2 diabetes.

Well established psychosocial correlates of weight management in type 2 diabetes have conceptual similarities to personality traits. Specifically, neuroticism and emotionality are related conceptually to emotional distress; conscientiousness to self-efficacy, and extraversion to both distress (negative association) and self-efficacy. However, there has been no comprehensive and direct investigation into the relative contributions of these constructs to weight management in adults with type 2 diabetes. What remains unanswered is whether personality domains, which are not fully represented by general and diabetes-specific well-being and self-efficacy, are important for understanding the experience of weight management among adults with type 2 diabetes.

The aim of this study was to examine the relationship between personality, general and diabetes-specific distress and self-efficacy and weight management behaviours (physical activity and healthy diet) and outcomes (body mass index; BMI) among Australian adults with type 2 diabetes. We hypothesised that: (1) conscientiousness and extraversion would be correlated positively with diabetes self-efficacy, (2) emotionality would be correlated positively, and extraversion correlated negatively, with diabetes distress, (3) conscientiousness and extraversion would have positive correlations with physical activity and healthy diet and a negative correlation with BMI, while emotionality would have negative correlations with physical activity and healthy diet and a positive correlation with BMI, and (4) conscientiousness, emotionality and extraversion would provide incremental explanation of variance for each weight management indicator over and above demographics, clinical characteristics, general and diabetes-specific wellbeing and self-efficacy.

Materials and methods

This study was a cross-sectional, Australia-wide, online survey conducted August to November 2019. Ethics approval was received from the Deakin University Human Research Ethics Committee (reference number 2019–152).

## Participants and recruitment

Eligible participants were adults (aged 18+ years), self-reporting a diagnosis of type 2 diabetes, and living in Australia (see Fig 1 for a participant flowchart). The study was advertised via the National Diabetes Services Scheme (NDSS), an Australian Government initiative administered by Diabetes Australia. An email invitation containing a link to the online survey was sent to 9,000 NDSS adult registrants with type 2 diabetes who had consented to being contacted for research purposes. The survey was also promoted on the websites and Facebook pages of Diabetes Victoria and the Australian Centre for Behavioural Research in Diabetes. Before completing the survey, all participants accessed a plain language statement and provided written informed consent. Prior to survey completion participants were informed that they would receive feedback on their personality responses and be eligible to win one of two AUD$200 gift vouchers. All participant data was deidentified after collection. Only the lead author had access to reidentified data for the purposes of assigning the gift vouchers.

## Measures

The online survey included demographic characteristics (e.g. age, gender) and clinical characteristics (e.g. diabetes treatment, duration, complications), including weight and height, used to calculate body mass index (BMI). Validated measures were used to assess physical activity and healthy diet, as well as personality domains and facets, general emotional well-being and self-efficacy, and diabetes-specific distress and self-efficacy (see Table 1).

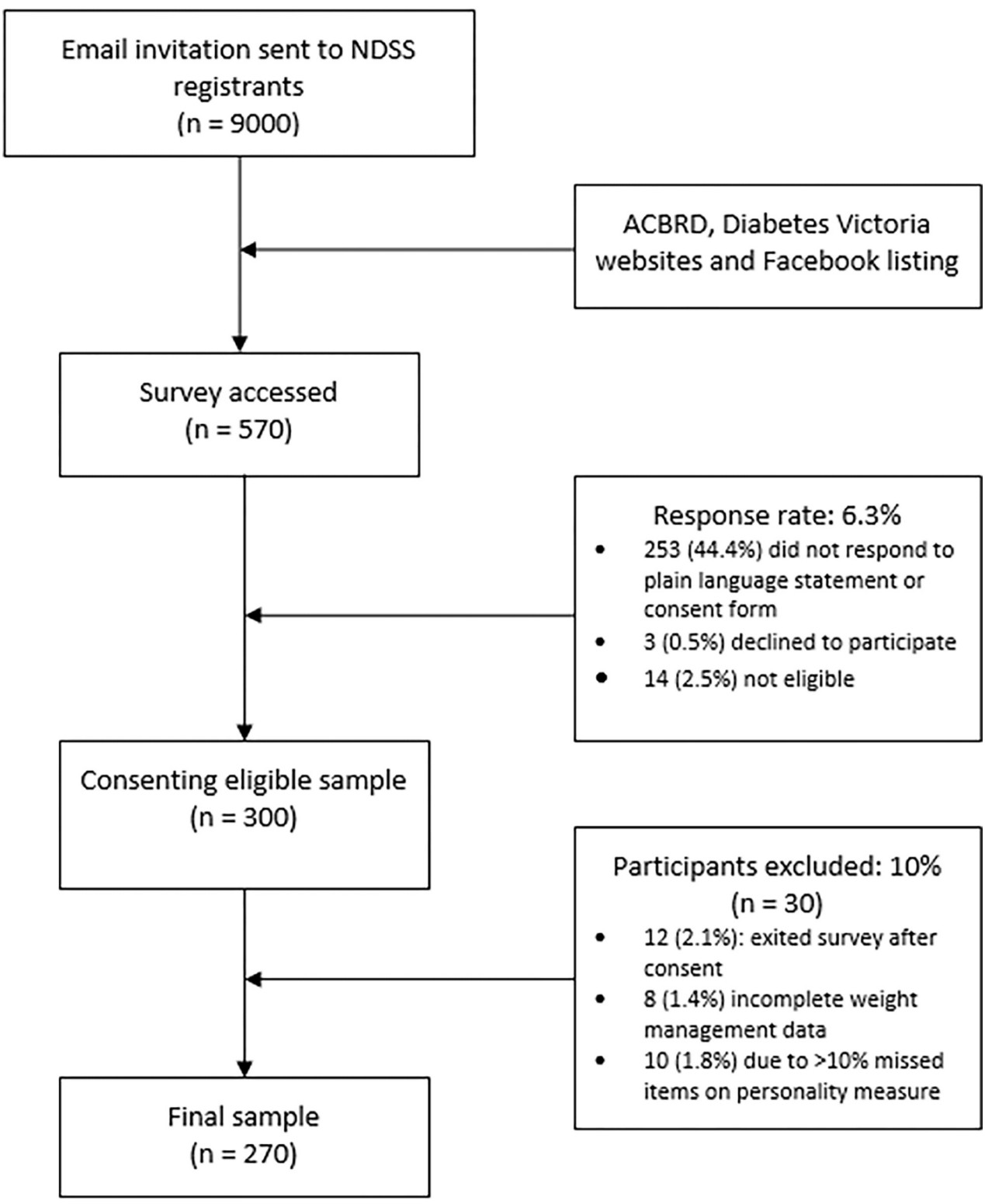

**Fig 1. Breakdown of response rate, exclusions and final sample for the study.**

**Table 1. Validated measures, including description, number of items, response format and scoring.**

| Concept | Measure | Description | Items | Response format and scoring |
|---|---|---|---|---|
| Personality | HEXACO-Personality Inventory-Revised | 6 broad personality domains (honesty-humility, emotionality, extraversion, agreeableness, conscientiousness, openness) and 25 narrow personality facets. In comparison to the Big Five structure, there is a slight reorganising of neuroticism (and renaming to emotionality) and extraversion. | 100 | 5-point scale from 1 = Strongly disagree to 5 = Strongly agree. Mean domain/facet scores (range: 1–5). Higher scores reflect higher levels of that domain. |
| General emotional well-being | WHO Well-Being Index (WHO-5) | Interest, engagement and mood over the preceding seven days | 5 | 6-point scale from 0 = At no time to 5 = All of the time. Total score (range: 0–25). Higher scores indicate better general emotional well-being. Scores <13 indicate depressive symptoms (likely depression) |
| Diabetes distress | Diabetes Distress Scale (DDS) | Extent to which aspects of diabetes have been experienced as distressing over the preceding month, rated across 4 subscales: Emotional burden, Physician-related distress, Regimen-related distress and Interpersonal distress. | 17 | 6-point scale from 1 = Not a problem to 6 = A very serious problem. Mean score (range: 1–6). Higher scores indicate greater diabetes distress. |
| General self-efficacy | General Self-Efficacy Scale (GSE) | Self-belief to cope with various difficult demands of life provided an overall score of general self-efficacy | 10 | 4-point scale from 1 = Not at all true to 4 = Exactly true. Total score (range: 10–40). Higher scores indicate greater general self-efficacy. |
| Diabetes self-efficacy | Diabetes Management Self-Efficacy Scale (DMSES) | Confidence to perform functional diabetes management behaviours across 4 subscales: Nutrition specific and weight, Nutrition general and medical treatment, Physical activity and Blood sugar. | 20 | 11-point scale, where 0 = "Cannot do at all", 5 = "Maybe yes, maybe no" and 10 = "Certainly can do". Total score (range: 20–200). Higher scores indicate greater self-efficacy in managing diabetes. |
| Body Mass Index (BMI) | Single items: height and weight | Body composition assessed as a ratio of weight to height to standardise weight class, e.g. obesity | 2 | Self-report weight (kg) and height (m) (formula: Weight / Height$^2$) |
| Physical activity | International Physical Activity Questionnaire–Short Form (IPAQ-SF) | Total physical activity (vigorous, moderate and walking activity) over the preceding week in MET-minutes (multiples of the resting metabolic rate). | 3 | Self-report of 10+ minutes of weekly vigorous, moderate and walking activity Measured in MET-minutes. Higher scores reflect greater physical activity. |
| Healthy diet | UK Diabetes Diet Questionnaire (UKDDQ) | Healthfulness of food consumed over the preceding month, across 3 subscales: Fibre, Saturated fat and Added sugar. | 24 | 6-point scale from 0 = Never or very rarely to 5 = Variable depending on food item (up to 4 + times a day. Total score (range: 0–120). Higher scores reflect healthier dietary choices. |

## Statistical analyses

Power analysis using G*power (version 3.1) showed that the study had 99.9% and 90% power to detect correlations of $r = .30$ and $r = .20$, respectively ($\alpha = .05$ two-tailed). Only participants with complete weight management behaviour and/or outcome data were included for analysis. Bivariate Pearson's correlations were calculated between personality and demographic, clinical, psychosocial variables (general wellbeing, general self-efficacy, diabetes distress and diabetes self-efficacy) and weight management indicators (physical activity, healthy diet and BMI). Multiple regression analyses were conducted to examine the contribution of personality to the explanation of variance in physical activity, healthy diet, and BMI, after adjusting for covariates. The linear regression blocks included (1) demographic and clinical variables, (2) general emotional well-being and general self-efficacy, (3) diabetes distress and diabetes self-efficacy, and (4) personality domains. The assumptions of multicollinearity, goodness of fit and independence pertaining to the linear regression models were tested. Tolerance statistics were assessed using the 1/variance inflation factor with criteria for review set at a <0.2 threshold.

The models were also tested for the influence and leverage of outliers using the Cook's distance statistic, with criteria for assessment of influence set at values >1. Exploratory regression analyses (S1 and S2 Tables) were also run, utilising reordered entry points for the explanatory variables across additional models whilst also introducing the diabetes distress subscales to examine suppressor effects. All statistical tests were two-sided and accepted as significant at $p < 0.05$. Data are available from the Open Science Framework project site.

## Results

Table 2 displays N = 270 (n = 30 excluded, refer to Fig 1) participants' demographic, clinical, psychosocial, and weight management characteristics. Participants had a mean±SD age of 61 ±12 years; 56% were women, and 75% were born in Australia. Participants reported a mean diabetes duration of 7.2±6.8 years, 86% percent had a BMI $\geq$25kg/m$^2$, and 14% reported using healthful diet and physical activity alone to manage their diabetes. Thirty seven percent of participants reported scores indicating depressive symptoms and likely depression [16] and 13% of scores indicated severe diabetes distress (total score >3.0).

Table 3 shows correlations between demographic, clinical, psychosocial, personality domains and weight management variables. S1 File and S3 Table describes the findings and displays correlations between personality facets, psychosocial and weight management variables.

### Correlates of personality: Demographic, clinical, and psychosocial characteristics

Among demographic and clinical characteristics, the strongest associations with personality were gender (females were higher on emotionality: $r$ = .30) and age (extraversion: $r$ = .25; emotionality $r$ = -.22). Personality domains were not significantly associated with diabetes duration or complications.

In general, personality domains had moderate-to-strong correlations with general emotional well-being (extraversion: $r$ = .66; emotionality: $r$ = -.37) and self-efficacy (extraversion: $r$ = .56; emotionality: $r$ = -.47). Personality traits also had moderate correlations to diabetes-specific distress (extraversion: $r$ = -.37; emotionality: $r$ = .31) and diabetes self-efficacy (extraversion: $r$ = .41; emotionality: $r$ = -.23; conscientiousness: $r$ = .21).

### Correlates of weight management indicators

The strongest correlations between personality and weight-management were observed between extraversion and physical activity ($r$ = .28), healthy diet ($r$ = .24) and BMI ($r$ = -.20). In addition, significant associations were observed for physical activity with conscientiousness ($r$ = .18) and openness ($r$ = .19), healthy diet with honesty-humility ($r$ = .18) and agreeableness ($r$ = .14) and BMI with emotionality ($r$ = .20).

General emotional well-being and diabetes self-efficacy showed moderate associations with physical activity (both $r$ = .42), BMI ($r$ = -.30 and -.40 respectively) and healthy diet ($r$ = .24 and .36 respectively) (Table 3). Significant correlations were observed between diabetes distress and healthy diet and BMI ($r$ = -.23 and .20 respectively). No significant relationship existed between diabetes distress and physical activity or between general self-efficacy and healthy diet and BMI.

### Regression models

The assumptions of multicollinearity, goodness of fit and independence pertaining to the linear regression models were met. No identified outliers returned a leverage value of

**Table 2. Participants' demographic, clinical, psychosocial and weight management characteristics (N = 270^).**

| Demographic variables | |
|---|---|
| Demographics | |
| Women, n (%) | 151 (56) |
| Age in years, mean ± SD | 60.9±11.6 |
| Relationship Status, n (%): Married/Defacto/Partnered | 177 (66) |
| Country of birth, n (%): Australia | 202 (75) |
| State of residence, n (%) | 7 (3) |
| Australian Capital Territory | 71 (26) |
| New South Wales | 2 (1) |
| Northern Territory | 53 (20) |
| Queensland | 17 (6) |
| South Australia | 9 (3) |
| Tasmania | 90 (33) |
| Victoria | 21 (8) |
| Western Australia | |
| Education, n (%) | |
| Completed Year 12 or less | 77 (29) |
| Trade training or diploma(s) | 77 (29) |
| University educated | 116 (42) |
| Employment, n (%) | |
| Employed | 100 (37) |
| Unemployed | 36 (14) |
| Retired | 124 (46) |
| Unpaid labor | 10 (3) |
| Annual household income, n (%) | |
| AUD$60,000+ | 114 (42) |
| Prefer not to say | 39 (15) |
| | |
| Clinical variables | |
| HbA1c >7%: n (%)^ | 77 (47) |
| Diabetes duration in years, mean±SD | 7.2±6.8 |
| Diabetes-related complications, n (%): >1* | 131 (49) |
| Diabetes treatments, n (%) | |
| Diet and physical activity | 156 (58) |
| Oral medication | 207 (77) |
| Insulin injections | 45 (17) |
| Non-insulin injections, e.g. Bydureon | 23 (9) |
| Other | 26 (10) |
| | |
| Psychosocial variables, median (IQR) | |
| General wellbeing: WHO-5 | 15 (9–19) |
| General self-efficacy: GSE | 31 (28–34) |
| Diabetes distress: DDS | 1.6 (1.2–2.4) |
| Diabetes self-efficacy: DMSES | 152 (124–169) |
| | |
| Weight management indicators | |
| Body Mass Index, n (%): kg/m2 | |
| <20 | 3 (1) |
| 20–24.9 | 36 (13) |
| 25–29.9 | 68 (25) |
| 30–34.9 | 72 (27) |

(*Continued*)

**Table 2.** (Continued)

| Demographic variables | |
|---|---|
| 35+ | 91 (34) |
| Physical activity (METmins): IPAQ-SF, median (IQR) | 1314 (396–2816) |
| Healthy diet: UKDDQ, mean±SD | 3.4±0.6 |

^HbA1c n = 165, Diabetes duration n = 266, General wellbeing n = 260, General self-efficacy n = 260, Diabetes distress n = 261, Diabetes self-efficacy n = 262, Physical activity n = 205, Healthy diet n = 264 (due to pairwise analyses).

*Diabetes-related complications included: heart disease/heart attack, kidney damage, neuropathy, retinopathy, sexual dysfunction, stroke and vascular disease.

significance. Prior to the addition of personality domains, the covariates (demographic, clinical, and psychosocial variables; model 3) accounted for 15–28% of the variance in the three weight management indicators (Table 4). With the addition of personality domains (model 4), the independent contribution of the covariates remained significant (except for diabetes duration), and for physical activity and healthy diet (but not BMI), personality explained an additional 3% of the variance, which was a statistically significant increase. Specifically, openness ($\beta = .16$) and emotionality ($\beta = .15$) added significantly to the explained variance in physical

**Table 3. Correlation matrix of weight management, personality domains, psychosocial, demographic and clinical variables (N = 270^).**

| Variable | 1 | 2 | 3 | 4 | 5 | 6 | 7 | 8 | 9 | 10 | 11 | 12 | 13 | 14 | 15 | 16 |
|---|---|---|---|---|---|---|---|---|---|---|---|---|---|---|---|---|
| Weight management | | | | | | | | | | | | | | | | |
| 1. Physical Activity | | | | | | | | | | | | | | | | |
| 2. Healthy Diet | .34** | | | | | | | | | | | | | | | |
| 3. Body Mass Index | -.33** | -.26** | | | | | | | | | | | | | | |
| Personality | | | | | | | | | | | | | | | | |
| 4. Honesty-Humility | .02 | .19** | .10 | | | | | | | | | | | | | |
| 5. Emotionality | -.06 | -.05 | .20** | .15* | | | | | | | | | | | | |
| 6. Extraversion | .28** | .24** | -.20** | -.12* | -.33** | | | | | | | | | | | |
| 7. Agreeableness | .06 | .14* | -.06 | .16* | -.03 | .34** | | | | | | | | | | |
| 8. Conscientiousness | .18* | .09 | -.10 | .07 | -.28** | .28** | -.02 | | | | | | | | | |
| 9. Openness | .19** | .04 | -.12 | .05 | -.11 | .27** | .14* | .05 | | | | | | | | |
| Psychosocial variables | | | | | | | | | | | | | | | | |
| 10. General Wellbeing | .42** | .24** | -.30** | -.10 | -.37** | .66** | .28** | .19** | .09 | | | | | | | |
| 11. General Self-Efficacy | .21** | .08 | -.08 | -.14* | -.47** | .56** | .17* | .39** | .22** | .51** | | | | | | |
| 12. Diabetes Distress | -.12 | -.23** | .20** | .04 | .31** | -.37** | -.19** | -.19** | .02 | -.53** | -.34** | | | | | |
| 13. Diabetes Self-Efficacy | .42** | .36** | -.40** | -.01 | -.23** | .41** | .19** | .21** | .14* | .56** | .40** | -.56** | | | | |
| Demographics | | | | | | | | | | | | | | | | |
| 14. Age | -.01 | .20** | -.25** | -.08 | -.22** | .25** | .09 | .04 | .05 | .30** | .09 | -.31** | .17* | | | |
| 15. Female gender | -.18* | -.05 | .35** | .20** | .30** | -.12* | .04 | -.10 | .10 | -.26** | -.18* | .32** | -.30** | -.24** | | |
| 16. Diabetes duration | -.04 | -.07 | -.06 | -.05 | .02 | .08 | .02 | -.06 | .08 | .07 | -.02 | .04 | .01 | .28** | -.10 | |
| 17. Complications | -.14* | -.10 | .18* | -.02 | -.01 | -.11 | -.02 | .02 | -.03 | -.07 | .00 | .10 | -.15* | .22** | -.04 | .17* |

Significant correlations: * = p<0.05, ** = p<0.01.

^ Physical activity n = 205, Healthy diet n = 264, General wellbeing n = 260, General self-efficacy n = 260, Diabetes distress n = 261, Diabetes self-efficacy n = 262, Diabetes duration n = 266 (due to pairwise analyses).

Diabetes-related complications include: heart disease/heart attack, kidney damage, neuropathy, retinopathy, sexual dysfunction, stroke and vascular disease.

**Table 4. Standardised coefficients of regression models explaining variance in weight management indicators (N = 266^).**

| | Physical Activity β* | Healthy Diet β* | BMI β* |
|---|---|---|---|
| Model 1 | | | |
| Age | -.02 | .26** | -.22** |
| Female gender | -.20** | .00 | .31** |
| Diabetes duration | -.03 | -.11 | -.01 |
| Diabetes comorbidities | -.14* | -.14* | .24** |
| Adjusted R² | .04* | .06** | .20** |
| Model 2 | | | |
| Age | -.14* | .20** | -.17** |
| Female gender | -.11 | .03 | .28** |
| Diabetes duration | -.03 | -.11 | .00 |
| Diabetes comorbidities | -.08 | -.11 | .20** |
| General emotional well-being | .45** | .23** | -.20** |
| General self-efficacy | -.03 | -.05 | .08 |
| Adjusted R² | .20** | .09** | .22** |
| Adjusted R² Change vs Model 1 | .16** | .03** | .02* |
| Model 3 | | | |
| Age | -.10 | .19** | -.20** |
| Female gender | -.11 | .09 | .26** |
| Diabetes duration | -.05 | -.11 | .01 |
| Diabetes comorbidities | -.06 | -.07 | .19** |
| General emotional well-being | .40** | .08 | -.12 |
| General self-efficacy | -.05 | -.10 | .12 |
| Diabetes distress | .29** | .02 | -.19* |
| Diabetes self-efficacy | .35** | .35** | -.34** |
| Adjusted R² | .28** | .15** | .28** |
| Adjusted R² Change vs Model 2 | .08** | .06** | .06** |
| Model 4 | | | |
| Age | -.09 | .19** | -.19** |
| Female gender | -.16* | .03 | .25** |
| Diabetes duration | -.07 | -.11 | .01 |
| Diabetes comorbidities | -.06 | -.05 | .19** |
| General emotional well-being | .45** | .01 | -.12 |
| General self-efficacy | -.07 | -.11 | .20** |
| Diabetes distress | .25** | .03 | -.18* |
| Diabetes self-efficacy | .30** | .33** | -.33** |
| Honesty-Humility | .05 | .20** | .06 |
| Emotionality | .15* | .04 | .07 |
| Extraversion | .00 | .19* | .02 |
| Agreeableness | -.07 | .00 | -.04 |
| Conscientiousness | .12 | .01 | -.08 |
| Openness | .16* | -.04 | -.11 |
| Adjusted R² | .31** | .19** | .29** |
| Adjusted R² Change vs Model 3 | .03* | .03* | .01 |

\* p<0.05 \*\* p<0.01.

^Physical activity, Model 1–4 n = 205, Healthy diet and BMI, Model 1 n = 266, Model 2–4 n = 260 (due to pairwise analyses).

activity, while honesty-humility ($\beta$ = .20,) and extraversion ($\beta$ = .19,) added significantly to the explained variance in healthy diet. In models 3 and 4, diabetes distress appears to be behaving as a suppressor variable (see also, S1 and S2 Tables), with a significant positive contribution to physical activity, while the bivariate correlations above shows these variables are not significantly associated ($r$ = -.12, p = .08).

## Discussion

Among adults with type 2 diabetes, the current study found that the personality domains of extraversion, conscientiousness and openness positively correlated with physical activity; extraversion, agreeableness and honesty-humility positively correlated with healthy diet; and extraversion (negatively) and emotionality (positively) correlated with BMI. Further, the personality domains of emotionality and openness as well as honesty-humility and extraversion explained additional unique variance (though small) in physical activity and healthy diet scores respectively, when adjusted for general and diabetes-specific distress and self-efficacy. However, contrary to the hypothesis, conscientiousness did not explain additional variance (over that explained by general and diabetes-specific emotional well-being and self-efficacy) in weight management indicators. None of the six personality domains provided unique explanation of variance in BMI. In addition, this study provides further evidence for the associations between personality domains (specifically, extraversion, emotionality, and conscientiousness) and general and diabetes-specific well-being and self-efficacy. As such, when considering the unique weight management challenges in type 2 diabetes, personality may help to broadly frame individual differences in the experience of general and diabetes-specific well-being and self-efficacy.

The correlations between personality and weight management indicators broadly corroborate previous research findings in general and obesity populations. For example, in this study, physical activity had the strongest positive correlation with extraversion, followed by openness and conscientiousness (both positive), among adults with type 2 diabetes. These findings are collectively consistent with those of large meta-analyses based on the general and obesity populations [14,15,17]. However, in previous type 2 diabetes studies in which the Big Five had been assessed, extraversion had not correlated with physical activity or other weight management indicators [5]. The activity and social facets of extraversion are common to physical activity behaviours [18] but have also been found to contribute to responder bias [19], which together may explain differences in findings. The current findings are also consistent with previous studies among people with type 2 diabetes that have identified positive correlations with conscientiousness [5]. Positive relationships between conscientiousness and physical activity are also common in the general and obesity populations [14,15,17]. The tendency toward higher levels of self-discipline and dutifulness that conscientious people embody may be responsible for self-regulating behaviour such as physical activity [5,17,18]. Our finding of a positive correlation between openness and physical activity, whilst not hypothesised, is supported by a meta-analysis of 88,400 participants, which found a small positive correlation between openness and physical activity in the general population [17]. A small positive correlation between openness and motor function has also been found in an overweight type 2 diabetes sample [20]. There are also longstanding links between information-seeking behaviour, characterised by higher levels of openness, and well-being which may apply to behaviours such as physical activity [21]. Of note, and contrary to the hypothesis, is our finding of no correlation between emotionality and physical activity. Whilst emotionality and neuroticism are not interchangeable, strong evidence exists for a negative relationship between neuroticism and physical activity [17]. The same body of research does not yet exist for HEXACO's emotionality and physical activity.

The observed positive correlations between healthy eating and extraversion, as hypothesized, and agreeableness (not hypothesised) are reflected in systematic reviews of general and obesity population studies [13,22]. However, in contrast to those published reviews [13,22], and among adults with type 2 diabetes [5], associations between healthful diet, conscientiousness and neuroticism (as represented by emotionality) were not identified in the current study, as hypothesised. Further, the observed positive correlation between honesty-humility and healthy diet (not hypothesised), was a first in a type 2 diabetes sample, but differed from a null finding in the general population [23]. Considering extraversion and healthy eating, it is well established that the social environment can impact eating behaviours [24], and this can be especially pronounced for people living with diabetes [25]. Negative associations regarding extraversion and eating habits do exist [26] and have not previously been demonstrated in a type 2 diabetes sample. As well as positive emotionality, much of the conceptual focus of extraversion relates to enjoyment of social interaction and relationships [27]. Nominally, the focus of honesty-humility concerns *how* we relate to others, with the strength of our relationships depending on our ability to defer self-interest to others, and research has shown how dispositional humility can affect social relationship quality [28]. The social and relational elements of eating may therefore point to an interaction between extraversion and honesty-humility's correlation with healthy diet, in which a person's social network plays an important role [29]. Although lower in incidence, systematic review demonstrates an overall positive association between eating behaviours and agreeableness [22], which may be linked with prosocial behaviour, as is the case with extraversion. In contrast to the hypothesis, but comparable to our physical activity findings, no correlation was observed between healthy eating and emotionality. Negative correlations between neuroticism (as it relates to emotionality) and healthy eating have previously been found in adults with type 2 diabetes [5] and the general and obesity populations [13,22]. However, research regarding negative emotions and eating have found coping strategies in response to negative emotions moderate eating behaviours [30]. This suggests that the assessment of an intermediary variable may have been able to further elucidate the relationship between emotionality and healthy eating in our study.

Personality and BMI correlations, as hypothesised, for extraversion (negative) and emotionality (positive), as it relates to neuroticism in our study, were representative of previous research among general, obesity and type 2 diabetes populations [5,13,31]. A notable difference in our study is the absence of the hypothesised negative BMI-conscientiousness correlation. With personality and weight management in type 2 diabetes being an under-researched area, the influence of conscientiousness with respect to BMI is more consistent and well researched in the general and obesity populations [13,31], but has been found in two type 2 diabetes studies [5]. Concerning our study's BMI-extraversion relationship, this is a first for a type 2 diabetes sample. Extraversion positively correlated with both physical activity and healthy diet, which together are more effective for long term weight loss [32]. Tailoring a type 2 diabetes intervention based, in part, on participants' introversion-extraversion stability also proved an effective method for reducing BMI and waist circumference [5]. It must be kept in mind that behaviours such as physical activity and healthy diet are proximal to the physiological mechanisms required to impact BMI, as well as glucose management independent of weight. Therefore, the pathway from behaviours to body weight changes can be affected by numerous environmental variables (e.g. neighborhood socioeconomic disparities [33]), which were not measured. Regarding neuroticism (as it relates to emotionality in this study), there is a strong body of evidence in the general and obesity populations [13,31,34] demonstrating its sub-optimal influence on BMI, demonstrated also in people with type 2 diabetes [5]. Whilst HEXACO's emotionality is less focused on negative emotions than the Big Five's neuroticism, the physiological impact of increased emotionality is well researched [35], and the role of

emotionality in the manifestation of weight gain may be a plausible mechanism for the present findings.

Diabetes distress and diabetes self-efficacy are the domain specific, or contextualised, adaptations of well-being and self-efficacy. Associations between variables are generally more likely where there is alignment of contextual criteria. With respect to weight and its role in type 2 diabetes, analysis of weight management is often contextualised through behaviours, e.g. physical activity and healthy eating. Given the items of the diabetes self-efficacy scale share more weight management-related behavioural context than the other psychosocial variables, the higher correlations seen between diabetes self-efficacy and weight management are not surprising. Understanding the relationships between weight management and general and diabetes specific well-being and self-efficacy, within a framework of individual differences, is the key value of personality measurement.

The regression modelling provides novel insight into personality and weight management with some support for the hypothesised relationships (emotionality-physical activity and extraversion-healthy diet). Testing the emotionality-physical activity finding for overlap with covariates, further exploratory analyses (S1 and S2 Tables) found diabetes distress, through the emotional burden subscale, acted as a suppressor for diabetes self-efficacy. This may indicate that lower levels of diabetes self-efficacy are associated with heightened emotional burden of participants striving to meet their physical activity requirements, theoretically leading to diabetes distress. The significance of openness in providing independent explanation of variance for physical activity may represent differences in the context of what openness and covariates measure and the relationship each has with being physically active, e.g. inquisitiveness (S3 Table). For healthy diet, whilst there is also conceptual overlap between extraversion and general emotional well-being, the social elements that extraversion measures may hold independent value for its explanation of variance in healthy eating not addressed by covariates. The association of honesty-humility and healthy eating may represent a similar mechanism in terms of the context of what is measured, e.g. fairness (S3 Table), and the relationship with healthy eating. An association between fairness and acceptance of interventions for healthier food choices has been previously demonstrated [36] and such interventions are a prominent feature of diabetes health promotion campaigns.

The personality-weight management literature within type 2 diabetes would benefit from additional, specific studies, to establish consistent patterns of association and longitudinal predictive data using objective measures. As the literature develops, implications for clinical practice from this study's findings, and others, may relate less to the time-constraints of general practice due to the comprehensive methods of robust personality assessment. For the person living with type 2 diabetes, a deeper understanding of how their personality influences their weight management efforts may be more appropriately addressed in a diabetes education setting. A better understanding of personality-weight management relationships may also allow for more informed diabetes management decisions in terms of matching the requirements of an intervention or course of action with the traits and personal characteristics needed to carry it out.

A major strength of our study is its design, being the first to evaluate personality, diabetes distress, diabetes self-efficacy and their associations with weight management. With respect to the sample, national representation across all states and territories of Australia was a key strength. Also, the sample's weight management indicator scores were similar to those reported in other type 2 diabetes samples with respect to activity levels [37] healthy diet scores [38] and proportion with overweight or obesity (90%) [1]. Regarding well-being, depressive symptoms in our sample were substantially higher than in the general Australian adults population (37% vs 10%) [39], but similar to other type 2 diabetes samples [40], and median self-

efficacy values were comparable to prior literature among the general population [41] and Australian adults with type 2 diabetes [42].

Several limitations should also be noted. First, whilst the use of the HEXACO personality inventory provides novel insights, direct comparisons with prior research, which has mostly used the Big Five, may have been limited. Most notably, HEXACO extraversion contains slightly more reversed neuroticism than Big Five extraversion and correlates more with measures of well-being [7], while neuroticism is distributed over honesty-humility, agreeableness and emotionality domains [43]. Second, the study used self-report measures for weight management indicators, which can be prone to responder bias, e.g. inflated physical activity self-reports [44], and did not collect adiposity measures, e.g. waist-to-hip ratio, which would have added to the robustness of the weight management indicators. Although other studies have found satisfactory self-report anthropometrics in overweight and obese samples [45,46]. Third, the diabetes-specific diet measure was constructed with a United Kingdom sample and modified for Australian use in this study, which may have affected responses. The validity and reliability of the IPAQ-SF in particular has been critiqued in previous research [47], and issues with data completeness in this study meant analyses involving physical activity, when compared to the *a priori* power calculations, were underpowered. Fourth, participants were self-selected, and the response rate was low (6.3%), though consistent with prior studies [40,48]. The sample was limited by a lack of cultural diversity, though broadly representative of the Australian population with diabetes in respect of age, but with a slightly lower prevalence of insulin use.

Overall, this study provides novel and comprehensive evidence of the cross-sectional relationships between personality and weight management behaviours and outcomes among adults with type 2 diabetes, in addition to the well-established demographic, clinical and psychosocial covariates. The findings enrich our understanding of how individual differences may influence the experience of type 2 diabetes and weight management. Whilst diabetes distress and diabetes self-efficacy share conceptual similarities with certain personality traits, this study suggests unique value in the measurement and consideration of personality as it relates to explained variance in physical activity and healthy diet.

## Supporting information

**S1 Checklist. STROBE statement—checklist of items that should be included in reports of *cross-sectional studies*.**
(DOCX)

**S1 Table. Regression model with cross-sectional explanatory variables for physical activity reorganised, demonstrating suppressor effect of diabetes distress.**
(DOCX)

**S2 Table. Regression model with cross-sectional explanatory variables for physical activity, reorganised utilising diabetes distress subscales to identify source of suppressor effect.**
(DOCX)

**S3 Table. Correlations between personality facets and weight management, general well-being and self-efficacy and diabetes-specific distress and self-efficacy.**
(DOCX)

**S1 File. Description of facet-level correlations.**
(DOCX)

**S1 Data.**
(DOCX)

## Acknowledgments

We thank the adults with type 2 diabetes who generously gave their time and participated in this study.

Participants in the study were recruited via the National Diabetes Services Scheme (NDSS.) The NDSS is an initiative of the Australian Government administered by Diabetes Australia.

## Author Contributions

**Conceptualization:** Ralph Geerling, Elizabeth Holmes-Truscott, Jane Speight.

**Data curation:** Ralph Geerling, Jeromy Anglim, Emily J. Kothe, Elizabeth Holmes-Truscott, Jane Speight.

**Formal analysis:** Ralph Geerling, Jeromy Anglim, Emily J. Kothe.

**Funding acquisition:** Ralph Geerling.

**Investigation:** Ralph Geerling.

**Methodology:** Ralph Geerling, Jeromy Anglim, Elizabeth Holmes-Truscott, Jane Speight.

**Project administration:** Ralph Geerling.

**Resources:** Ralph Geerling.

**Supervision:** Jeromy Anglim, Emily J. Kothe, Elizabeth Holmes-Truscott, Jane Speight.

**Validation:** Ralph Geerling.

**Writing – original draft:** Ralph Geerling.

**Writing – review & editing:** Ralph Geerling, Jeromy Anglim, Emily J. Kothe, Miranda T. Schram, Elizabeth Holmes-Truscott, Jane Speight.

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
