## [Decision Letter · Decision Letter 0]

10 May 2023

PONE-D-23-08813Relationships between personality, emotional well-being, self-efficacy and weight management among adults with type 2 diabetes: results from a cross-sectional surveyPLOS ONE

Dear Dr. Geerling,

Thank you for submitting your manuscript to PLOS ONE. After careful consideration, we feel that it has merit but does not fully meet PLOS ONE’s publication criteria as it currently stands. Therefore, we invite you to submit a revised version of the manuscript that addresses the points raised during the review process.

 In the revision, please address all comments by reviewer #1 regarding the abstract, introduction (the rationale and provide a more detailed explanation of used terms where needed), discussion and the analysis. Also, revise the grammar and formatting where needed.

We look forward to receiving your revised manuscript.

Kind regards,

Srebrenka Letina, Ph.D.

Academic Editor

PLOS ONE

Journal Requirements:

Additional Editor Comments:

A minor revision is requested.

Reviewers' comments:

Reviewer's Responses to Questions

**Comments to the Author**

1. Is the manuscript technically sound, and do the data support the conclusions?

Reviewer #1: Yes

Reviewer #2: Yes

2. Has the statistical analysis been performed appropriately and rigorously? 

Reviewer #1: Yes

Reviewer #2: Yes

3. Have the authors made all data underlying the findings in their manuscript fully available?

Reviewer #1: Yes

Reviewer #2: Yes

4. Is the manuscript presented in an intelligible fashion and written in standard English?

Reviewer #1: Yes

Reviewer #2: Yes

5. Review Comments to the Author

Reviewer #1: The article provides insights into the relationships between personality, emotionality, wellbeing, self-efficacy and weight management in adults living with type 2 diabetes.

Overall the article would benefit from further reading in the area and citing more studies and systematic reviews from the area of weight management and/or type 2 diabetes weight management rather than general population research. The article reports there is a lack of research in this area but there is research into these factors within weight management specifically related to emotionality/regulation, self-efficacy and "healthful behaviours". Throughout terms are used with little explanation (e.g. health-harming cycles (where is the evidence if they are not proactive in PA/healthy eating that they engage in health-harming behaviour), or directionality (e.g. when referring to healthy eating or PA, is this an increase or decrease?). Specific points listed below:

Abstract - The abstract lacks the typical format (i.e. the intro) and should pull out key findings for the reader.

Introduction - The introduction needs to strengthen the rationale. What does the weight management (ideally in type 2 diabetes) tell us, what is missing and how the research adds to this.

Methods - A couple of queries which need clarified in the text - was the study across all of Australia or just Victoria region? Were participants emailed a link and/or complete the survey via the online advert?

Results - It would be useful at the start to state how many participants took part/were excluded. Was data collected on where they were from so there's insight into what areas of Australia were represented? In some of the tables the number of participants varied - why is this? In the regression models please plainly state the variance explained for model 4.

Discussion - did everything correlate with increased PA/healthy eating? The directionality needs to be stated. The comment on extraversion/social interactions/emotionality requires a reference. Limits/Strengths are thorough. Possibly of interest -there is some evidence self-reported measures of weight can be quite accurate. A strength is also the nature of the study - collecting data at population level! The discussion would benefit from clearer explanations of the implications of these findings for weight management in type 2 diabetes and directions for future research.

Reviewer #2: This is a very informative publication that addresses a gap in the field regarding personality traits and weight management in adults with Type 2 diabetes. Overall there are some grammatical and formatting changes that are necessary to improve readability but this paper will advance the field of study.

6. PLOS authors have the option to publish the peer review history of their article (what does this mean?). If published, this will include your full peer review and any attached files.

Reviewer #1: **Yes: **Meigan Thomson

Reviewer #2: No

---

## [Author Response · Author response to Decision Letter 0]

7 Jul 2023

7 July 2023

Dr Srebrenka Letina

Academic Editor

PLOS ONE

Dear Dr Letina

Re: Manuscript ID: PONE-D-23-08813 Relationships between personality, emotional well-being, self-efficacy and weight management among adults with type 2 diabetes: results from a cross-sectional survey

Thank you for considering our manuscript for publication. We have read the editorial and reviewer feedback with interest and respond to each point, in turn, below. 

Where appropriate, we have also revised our manuscript using bold and coloured text, plus a cleaned version, as requested. 

We believe these revisions have strengthened the manuscript and thank you for the opportunity to do so.

We look forward to your response in due course.

Kind regards

Ralph Geerling

 

Academic editor comments:

Thank you for submitting your manuscript to PLOS ONE. After careful consideration, we feel that it has merit but does not fully meet PLOS ONE’s publication criteria as it currently stands. Therefore, we invite you to submit a revised version of the manuscript that addresses the points raised during the review process.

Authors’ response: Thank you. We provide responses to the comments below.

Reviewer comments:

Reviewer: 1

Comments to the Author

The article provides insights into the relationships between personality, emotionality, wellbeing, self-efficacy and weight management in adults living with type 2 diabetes.

Authors’ response: Thank you for your time and expertise in reviewing our manuscript. We have attended to your suggestions and believe these have strengthened the manuscript. We trust these changes will be satisfactory. 

Overall the article would benefit from further reading in the area and citing more studies and systematic reviews from the area of weight management and/or type 2 diabetes weight management rather than general population research. The article reports there is a lack of research in this area but there is research into these factors within weight management specifically related to emotionality/regulation, self-efficacy and "healthful behaviours". 

Authors’ response: Thank you for this suggestion. We have added further wording and references to bolster the general population findings with additional obesity and type 2 diabetes studies and meta-analyses/systematic reviews where relevant throughout the introduction and discussion. 

Throughout terms are used with little explanation (e.g. health-harming cycles (where is the evidence if they are not proactive in PA/healthy eating that they engage in health-harming behaviour), or directionality (e.g. when referring to healthy eating or PA, is this an increase or decrease?).

Authors’ response: Thank you for highlighting the problematic nature of this language. Health-harming and similar 

wording have now been removed from the manuscript. 

Abstract

The abstract lacks the typical format (i.e. the intro) and should pull out key findings for the reader. 

Authors’ response: Thank you for pointing this out. Unfortunately, the typical format for the abstract does not form part of the PLOS ONE guidelines for the manuscript so we have been unable to make this change. 

Introduction

The introduction needs to strengthen the rationale. What does the weight management (ideally in type 2 diabetes) tell us, what is missing and how the research adds to this. 

Authors’ response: Thank you for this suggestion. We have now included additional wording (Intro para 4) to strengthen the rationale and hope it meets the expectations for clarifying for the reader why weight management is being addressed.

Methods

A couple of queries which need clarified in the text - was the study across all of Australia or just Victoria region? Were participants emailed a link and/or complete the survey via the online advert? 

Authors’ response: Thank you for highlighting this. We have now clarified that this was an Australia-wide survey and that participants received a link to the online survey in the email they received (Materials and methods paras 1 and 2). 

Results

It would be useful at the start to state how many participants took part/were excluded. Was data collected on where they were from so there's insight into what areas of Australia were represented? In some of the tables the number of participants varied - why is this? In the regression models please plainly state the variance explained for model 4. 

Authors’ response: Thank you for pointing this out. We have added up front (Results para 1) the participants included and excluded. We have also added the breakdown of participants across the states and territories in Table 2. The reason for the differing number of participants has been added under each table (due to pairwise analyses). Additional wording has also been added in the Regression Models para to plainly state the explained variance for model 4. 

Discussion

did everything correlate with increased PA/healthy eating? The directionality needs to be stated. The comment on extraversion/social interactions/emotionality requires a reference. Limits/Strengths are thorough. Possibly of interest -there is some evidence self-reported measures of weight can be quite accurate. A strength is also the nature of the study - collecting data at population level! The discussion would benefit from clearer explanations of the implications of these findings for weight management in type 2 diabetes and directions for future research. 

Authors’ response: Thank you for these suggestions. The directionality of the correlations has now been added throughout the manuscript. A reference has now been added (Discussion para 3) for the emotionality/extraversion/social interaction comment. Within the strengths and limitations paragraphs we have added wording regarding the survey being national, and wording and citations regarding self-report accuracy. A separate paragraph has also been added regarding directions for future research and clinical implications. 

Reviewer: 2

Comments to the Author

This is a very informative publication that addresses a gap in the field regarding personality traits and weight management in adults with Type 2 diabetes. Overall there are some grammatical and formatting changes that are necessary to improve readability but this paper will advance the field of study. 

Authors’ response: Thank you for your time and expertise in reviewing our manuscript. We trust that the changes made to the manuscript overall, address the issues raised concerning readability.

---

## [Decision Letter · Decision Letter 1]

25 Sep 2023

Relationships between personality, emotional well-being, self-efficacy and weight management among adults with type 2 diabetes: results from a cross-sectional survey

PONE-D-23-08813R1

Dear Dr. Geerling,

We’re pleased to inform you that your manuscript has been judged scientifically suitable for publication and will be formally accepted for publication once it meets all outstanding technical requirements.

Kind regards,

Srebrenka Letina, Ph.D.

Academic Editor

PLOS ONE

Additional Editor Comments (optional):

Reviewers' comments:

Reviewer's Responses to Questions

**Comments to the Author**

1. If the authors have adequately addressed your comments raised in a previous round of review and you feel that this manuscript is now acceptable for publication, you may indicate that here to bypass the “Comments to the Author” section, enter your conflict of interest statement in the “Confidential to Editor” section, and submit your "Accept" recommendation.

Reviewer #1: All comments have been addressed

Reviewer #3: All comments have been addressed

2. Is the manuscript technically sound, and do the data support the conclusions?

Reviewer #1: Yes

Reviewer #3: Yes

3. Has the statistical analysis been performed appropriately and rigorously? 

Reviewer #1: Yes

Reviewer #3: Yes

4. Have the authors made all data underlying the findings in their manuscript fully available?

Reviewer #1: Yes

Reviewer #3: Yes

5. Is the manuscript presented in an intelligible fashion and written in standard English?

Reviewer #1: Yes

Reviewer #3: Yes

6. Review Comments to the Author

Reviewer #1: Thank you for addressing the previous comments, the article is now much clearer and reflects wider reader and consideration of the wider evidence in the field.

Reviewer #3: This is an interesting and important study. Based on my review of the responses to reviewers and the tracked changes I believe that this meets the requirements to be accepted for publication.

7. PLOS authors have the option to publish the peer review history of their article (what does this mean?). If published, this will include your full peer review and any attached files.

Reviewer #1: No

Reviewer #3: **Yes: **Emma Berry

---

## [Editor Report · Acceptance letter]

20 Oct 2023

PONE-D-23-08813R1 

Relationships between personality, emotional well-being, self-efficacy and weight management among adults with type 2 diabetes: results from a cross-sectional survey 

Dear Dr. Geerling:

I'm pleased to inform you that your manuscript has been deemed suitable for publication in PLOS ONE. Congratulations! Your manuscript is now with our production department. 

Kind regards, 

on behalf of

Dr. Srebrenka Letina 

Academic Editor

PLOS ONE